# The Relationship between Rainfall Pattern and Epilithic Diatoms in Four Streams of Central-Western Korea for Three Years (2013–2015)

**DOI:** 10.3390/ijerph20054099

**Published:** 2023-02-24

**Authors:** Eun-A Hwang, In-Hwan Cho, Ha-Kyung Kim, Chen Yi, Baik-Ho Kim

**Affiliations:** 1Department of Environmental Science, Hanyang University, Seoul 04763, Republic of Korea; 2Department of Life Science and Research Institute for Natural Sciences, Hanyang University, Seoul 04763, Republic of Korea

**Keywords:** diatoms, rainfall pattern, land use, soil characteristics, community dynamic index

## Abstract

To study the effect of rainfall patterns on diatom communities in four major central western streams on the Korean Peninsula during the monsoon seasons of 2013 through 2015, we measured precipitation, environmental factors, and epilithic diatoms at 42 sites before (May) and after (August and September) each monsoon. The Mangyeonggang river and Sapgyocheon stream (SS) had a high percentage of low-permeability soil, and the stream had the highest proportion (49.1%) of surrounding land in urban areas. Precipitation and precipitation frequency was closely correlated with electrical conductivity and nutrients, and this was particularly evident in SS. Epilithic diatom abundance for the most abundant species as, *Navicula minima*, decreased in the stream in 2013 and 2014 and increased in 2015 when precipitation and precipitation frequency were low. This was not clearly distinguishable in the ecological characteristics of each watercourse’s indicator species, except in SS. The dynamic community index was highest in 2015 (ca. 5.50), and the annual changes in the index were clearly shown in SS. The precipitation pattern and the dynamic community index were negatively correlated (r = −0.026~−0.385), and the precipitation within 2 weeks (r = −0.480 for SS) before the second sampling and the frequency of 10 mm of precipitation were closely correlated in the stream (r = −0.450 for SS). The distribution of epilithic diatoms in the four watercourses is therefore affected by monsoon precipitation and precipitation frequency, and the dynamic community index is determined by soil characteristics and land use.

## 1. Introduction

A monsoon is a seasonal weather phenomenon in which wind direction changes due to a temperature difference between the continent and the surrounding ocean [1]. South Korea is greatly affected by such phenomena as it is a peninsula, resulting in intensive rainfall during the summer [2,3]. The country’s average annual precipitation is 1274 mm, which is 1.3 times that of the global average (973 mm), but more than 50% of the precipitation falls in the summer rainy season due to the effect of monsoons [4]. However, unpredictable torrential rains, droughts, and heat waves due to climate change, and global warming in particular in recent years [5], affect rainfall patterns during the monsoon season. According to recent studies, an environmental condition that hinders rainfall generation due to an increase in average temperature can decrease rainfall frequency and increase droughts and evaporation [6,7,8].

The Geumgang River, Mangyeonggang River, Dongjingang River, and Sapgyocheon stream are classified as major watercourses that flow westward in the mid-western region of the Korean Peninsula [9,10]. Population increases and industrial development in recent years have changed the watershed environment, and industrial wastewater and urban sewage now pollute the waters [11,12,13]. A rapid change in water quality follows rainfall, floods, and intensive rainfalls during the monsoon period [14], and the pollution loads in downstream areas are largely due to the influx of surface water and tributaries [15].

Rainfall can change a river’s physiochemical characteristics, such as flow rate, flow velocity, turbidity, and nutritional substances [16,17,18], and affect aquatic organisms, including epilithic diatoms, fish, and benthic macroinvertebrates, altering species distribution and community variation [19,20,21]. Epilithic diatoms live by attaching themselves to substrates such as stones, branches, and aquatic organisms [22]. They are small in size, and the standing crop is rich [23,24], making them primary producers in aquatic ecosystems [25,26]. Their mobility is low, and they are sensitive to physical-chemical changes such as water temperature, light intensity, turbidity, nutrient supply, flow velocity, and flow rate, making them ideal indicators of long-term environmental change [27]. The epilithic diatom community can be used to identify variations in aquatic ecosystems as they react to rapid changes in water temperature, nutrients, and flow rate, which can be affected by monsoon rainfall [28,29,30]. The relationship between rainfall patterns and an epilithic diatom community can help identify the effects of intensive monsoon rainfalls on stream environments and biotic communities [31].

Studies have been conducted that monsoon affects changes in water quality and fish species on the Korean Peninsula [4,32,33,34]. However, few studies on epilithic diatoms have been carried out. In the case of rapid changes in physical and chemical factors, epilithic diatoms are appropriate indicator species because their movement is limited, and accumulated environmental changes are more evident in diatoms when compared with water quality, which shows only short-term results, or fishes whose habitat range is wide [35]. According to a recent study, rainfall affects the variation of the diatom community [31,35]. Studies of the population dynamics of epilithic diatoms and community variation are needed to identify the biological and environmental changes due to monsoon rainfalls in rivers in the central and western part of the Korean Peninsula and the biological and environmental changes associated with monsoon rainfall patterns that may be affected by global warming.

The present study describes the effect of rainfall patterns during the monsoon period on epilithic diatom communities. Environmental factors and epilithic diatoms in the Geumgang River, Mangyeonggang River, Dongjingang River, and Sapgyocheon stream in the mid-western region of the Korean Peninsula were surveyed before and after monsoons in 2013, 2014, and 2015.

## 2. Materials and Methods

### 2.1. Survey Sites

The Geumgang River (GR) is one of three major rivers in South Korea. Its main channel is 395.9 km long, flowing to the Yellow Sea via Daejeon, Sejong, and Nonsan. It has 31 tributaries, including the Bocheong-Cheon, Namdae-Cheon, and Gap-Cheon. The Mangyeonggang River (MR) and Dongjingang River (DR), which are in the south of the mid-western watershed, are mid-sized rivers that are joined by many tributaries and flow into the western sea across the Honam Plains. Around the Sapgyocheon stream (SS) in the north, a wide alluvial land, which is the sedimentary terrain of the river, has developed, and small and mid-sized cities and industrial complexes have been built around the stream. In the mid- to downstream reaches, agricultural land is leading [36,37,38,39,40,41], and the proportion of agricultural land is relatively high compared with areas around other watercourses due to a downstream reclamation project [11,16].

From 2013 to 2015, a survey was conducted at 42 sites on 4 major rivers in the mid-western region of the Korean Peninsula: 10 sites on the GR, 9 on the MR, 11 on the DR, and 12 on SS (Figure 1). The survey avoided periods of intense rainfall. The first survey was conducted in May, before the rainfall season, and the second was conducted in September, after the rainfall season, of each year, for a total of six surveys to investigate the water environment and epilithic diatoms (Figure 2).

### 2.2. Rainfall Patterns

To identify the rainfall pattern in major rivers in the mid-western region of the Korean Peninsula due to the influence of monsoons, a water environment information system was used (Figure 2). The rainfall amount and frequency determined the rainfall pattern. The rainfall amount was calculated by summing all rainfall over a certain period before the second sampling, and the rainfall frequency was the number of rainfall events equal to or exceeding a certain amount of precipitation during the first and second sampling periods.

### 2.3. Environmental Factors

The field survey table, including river width, water depth, and land use, identified the topography and river characteristics of the survey sites. To create the land use map around the survey sites for each river, a radius of 50 m from the sampling place was divided into urban, forest, agriculture, industrial complex, and livestock categories. To identify the soil characteristics in the survey sites, a radius of 50 m from the sampling place was categorized into A, B, C, and D classes (Table 1) based on data from the Regional Land Management Office [12,13,14,15].

To investigate the water quality in the survey sites, water temperature, dissolved oxygen (DO), pH, electrical conductivity, and turbidity were directly measured in the field using the Horiba U-50 mobile water quality meter. In order to measure biological oxygen demand (BOD), a 300 mL sample of water collected at each site was put into a Winkler bottle and moved to a laboratory in cold and dark conditions and then incubated for five days at 20°C under a light-blocking condition according to Winkler’s azide method. BOD was the difference between the amount of consumed dissolved oxygen (DO) and the measured concentration of DO at the site. In order to analyze the water quality of total nitrogen (TN), ammonia nitrogen (NH_3_-N), nitrate nitrogen (NO_3_-N), total phosphorus (TP), and phosphorus phosphate (PO_4_-P), the water in the site was sampled in a sterile 2 L water sampling bottle and transported to a laboratory in cold and dark conditions. Measurements of TN, NH_3_-N, NO_3_-N, TP, and PO_4_-P were taken using cadmium reduction, colorimetry, cadmium reduction, the ascorbic acid method after persulfate decomposition, and the ascorbic acid method, respectively [42].

### 2.4. Epilithic Diatoms

To collect epilithic diatoms, a site with flowing water that was submerged for more than three to four weeks was chosen. Wide and flat natural rocks whose upper side and water flow were in the same direction were selected. A soft brush was used to scrub diatoms from the upper part (5 × 5 cm^2^) of the sampled substrate, and the samples were then cleansed with 200 mL of distilled water, fixed in Lugol’s solution, and transported to the laboratory. Permanent samples for taxonomic identification of the diatoms were prepared using a Naphrax encapsulant (Brunel Microscopes Ltd., Chippenham, UK) after acid treatment and washing with a KMnO_4_ catalyst using the permanganate method [43]. To accurately survey the species composition of the diatoms, an optical microscope (Nikon E600, Tokyo, Japan) under 400× to 1000× magnification (Figure 3). And the Krammer and Lange-Bertalot, and Patrick and Reimer references [44,45,46] were used. In order to determine the population size of the sampled diatoms (in cells/cm²), live diatoms in the fixed specimens were counted using a San Rafael chamber. The relative abundance in each sample was calculated, and the density of epilithic diatoms was calculated by multiplying the population size in a unit area by their relative abundance.

### 2.5. Indicator Species Analysis

In order to identify epilithic diatoms in the rivers of the mid-western region of the Korean Peninsula, indicator species analysis (ISA) was conducted [47]. The indicator value (IndVal) is derived from the relative abundance and cell density of epilithic diatoms. The IndVal has a range of 0 to 100, and a high index value means that the index property is large [48]. In this study, when the IndVal was ≥10, species with an IndVal five times higher than that of another group were selected as the indicator species for that group.

### 2.6. Community Dynamic Index

A dynamic community index (CDI) was calculated to identify community variation of epilithic diatoms due to rainfall. Principal component analysis (PCA) was conducted, and the distribution of each site was drawn in a two-dimensional space. The average community variation was calculated by computing the distance between the first and second surveys at each site [31].
(1)$$d_{i}=\sqrt{{\left(x_{i}-x_{i}^{\prime }\right)}^{2}+{\left(y_{i}-y_{i}^{\prime }\right)}^{2}}$$
d*_i_*: the distance between two communities with different times at the specific site in the PCA dimension. x*_i_*, y*_i_*: the first sampling site in PCA dimension before monsoon. x*_i_*^′^, y*_i_*^′^: the second sampling site in the PCA dimension after the monsoon.
(2)$$\mathrm{CDI}\;=(\overset{n}{\underset{i=1}{\sum }}d_{i})/n$$
d*_i_*: the distance between two communities with different times at the specific site in the PCA dimension. *n*: the number of sampling sites or communities.

### 2.7. Data Analysis

In order to compare the physical-chemical environmental and biological factors (number of species, abundance) and community indexes between the groups, analysis of variance (ANOVA) was applied. Differences in the rivers before and after the monsoon were subjected to a *t*-test. A Pearson correlation analysis and SPSS software were used to identify correlations between rainfall patterns, environmental factors, and epilithic diatom communities. In other statistical analyzes, data were converted into log10 values to increase the accuracy of the analysis by reducing the variation of epilithic diatoms and environmental factors. To prevent an error in log values when data with a value of 0 were converted, 1 was added to each variable to calculate Log10 (n + 1). PC-ORD software was used to calculate the CDI.

## 3. Results and Discussion

### 3.1. Rainfall Patterns

The rainfall amount during the monsoon period was the largest in 2013 and the lowest in 2015 in all rivers (Table 2). The rainfall pattern sampled for a month was different for each river. The respective sums of rainfall in each river a week before the second sampling in 2013, 2104, and 2015 were 10.0 mm, 77.0 mm, and 17.6 mm for the Geumgang River; 14.6 mm, 33.7 mm, and 11.7 mm for the Mangyeonggang River; and 0.3 mm, 97.9 mm, and 18.7 mm for the Dongjingang River, which showed the highest rainfall in 2014. However, the Sapgyocheon stream had 70.7 mm, 55.5 mm, and 6.5 mm of rainfall in 2013, 2014, and 2015, respectively. This was the same pattern as that of the monsoon rainfall.

The rainfall frequency over a certain intensity during the monsoon period was counted. The frequency of 10 mm or more of precipitation was the highest in 2013 and the lowest in 2015 in the Geumgang River, but the frequency of rainfall of at least 30 mm was the highest in 2014. For the Mangyeonggang River and Dongjingang River, the frequency of rainfall ≥10 mm was the highest in 2014. The frequency of rainfall ≥30 mm was highest in 2013 and the lowest in 2015, similar to the rainfall pattern for the monsoon period. For the Sapgyocheon stream, the frequencies of rainfall of all intensities were the highest in 2013 and the lowest in 2015. Both the rainfall amount and frequency in the Sapgyocheon stream were the highest in 2013 and the lowest in 2015, as was the case with the rainfall patterns during the monsoon period.

### 3.2. Land Use and Soil Characteristics

The land use map showed a distinctive difference from river to river (Figure 4). The proportion of land devoted to agriculture was high around the Geumgang River at 52% and around the Dongjingang River at 60% (ANOVA, *p* < 0.01), while the proportion devoted to urban development was relatively high in the Sapgyocheon stream at 54.6% (ANOVA, *p* < 0.01). In the Mangyeonggang River, the proportions of urban and forested areas were high at 44.1% and 23.5% (ANOVA, *p* < 0.05 for all), respectively. Land use around the rivers had a significant impact on water quality [49]. Nutrients and pollution were lower in forested areas as forests can naturally purify water quality [50]. However, the agricultural and urban areas had high levels of pollution compared with the forested areas due to the effects of non–point-source pollution [51]. In particular, pollution was reportedly high due to an impermeability layer and commercial and industrial zones in urban areas [19].

Soil characteristics by the rivers were also surveyed (Figure 5). There was no significant difference in the proportion of class A, B, and C soils in the rivers, but the proportions of class D soil were lower in the Geumgang River (10.0%) and the Dongjingang River (9.1%) and higher in the Mangyeonggang River (53.3%) and Sapgyocheon stream (49.1%) (ANOVA, *p* < 0.01). Soil characteristic is an important factor in analyzing runoff due to rainfall. For example, Class D soil causes relatively higher runoff [13].

### 3.3. Environmental Characteristics

The environmental factors in the four studied rivers showed distinctive differences before and after monsoons (Table 3). The water temperature increased significantly from 20.8 °C to 24.4 °C in the Geumgang River, from 21.2 °C to 25.1 °C in the Mangyeonggang River, from 19.9 °C to 25.1 °C in the Dongjingang River, and from 21.2 °C to 24.7 °C in Sapgyocheon stream before and after monsoons (ANOVA, *p* < 0.01). Before the monsoon, the Sapgyocheon stream had clearly higher DO levels (8.2 mg/L to 6.9 mg/L) compared with that of the Geumgang (7.3 mg/L to 7.8 mg/L), Mangyeonggang (7.3 mg/L to 7.9 mg/L), and Dongjingang (7.7 mg/L to 8.3 mg/L) rivers (ANOVA, *p* < 0.05). After the monsoons, DO was significantly reduced in the Sapgyocheon stream, reaching its lowest value (*t*-test, *p* < 0.01, ANOVA, *p* < 0.01).

Electrical conductivity increased in all rivers before and after monsoons, but it varied from river to river. Electrical conductivity was relatively low in the Geumgang River (287.9 μS/Cm; 397.0 μS/Cm) and the Dongjingang River (191.3 μS/Cm; 288.2 μS/Cm), whereas it was relatively high in the Mangyeonggang River (342.2 μS/Cm; 530.0 μS/Cm) and Sapgyocheon stream (420.1 μS/Cm; 498.7 μS/Cm) (ANOVA, *p* < 0.05). It increased in all rivers after monsoons.

Turbidity levels decreased in all rivers and were the highest in the Geumgang River (84.2 nephelometric turbidity units [NTU]; 67.5 NTU) before and after the monsoon. TN levels were the highest in the Sapgyocheon stream (4.37 mg/L; 2.95 mg/L) before and after the monsoon (ANOVA, *p* < 0.01), but no significant difference was seen in the Geumgang River (2.72 mg/L; 2.51 mg/L), Mangyeonggang River (2.63 mg/L; 2.35 mg/L), and Dongjingang River (2.12 mg/L; 1.97 mg/L). TN levels decreased in all rivers after the monsoon, particularly in the Sapgyocheon stream (*t*-test, *p* < 0.05). NH_3_-N was also the highest in the Sapgyocheon stream (1.79; 0.80) before and after the monsoons, whereas no significant difference was found in the Geumgang River (0.32; 0.39), Mangyeonggang River (0.75; 0.31), and Dongjingang River (0.06; 0.07). TP levels before the monsoon were the highest in the Sapgyocheon stream, and no significant change was found in any rivers before and after the monsoon.

The environmental factors in the rivers differed significantly depending on the sampling period (Figure 6). Water temperatures increased after monsoons, particularly in 2015, when the rainfall amount and frequency were small. Electrical conductivity increased distinctively in 2013 when the rainfall amount and frequency were large and decreased in 2014 and 2015 after the monsoons when the rainfall amount and frequency were small. Rainfall in the monsoon periods causes rain runoff, introducing pollution sources into the rivers [51]. Urban and agricultural areas, in particular, where the soil permeability is lower than that of the wetlands and forests, had a large amount of soil and particulate-matter runoff, intensifying point and non-point pollution sources and increasing nutrient runoff [52,53]. The sampling sites were located in the mid-to-downstream reaches of the rivers and included many agricultural and urban areas with multiple pollution sources.

Variation in electrical conductivity in 2013 was high, with rich nutrients, compared with that of other years. TN, NH_3_-N, and TP were higher in the Sapgyocheon stream but low in the Geumgang River, Mangyeonggang River, and Dongjingang River, with fewer nutrients. The variation range in the Sapgyocheon stream was clear before and after monsoons. In particular, TN and NH_3_-N decreased more dramatically in 2015 when the rainfall amount and frequency were larger than in 2013. The influx of pollution sources was closely related to runoff from rainfall and surrounding land [54,55]. Soil with low permeability caused more runoff [13]. Sapgyocheon stream had a higher concentration of nutrients and a higher proportion of soil with low permeability compared with the three rivers.

The annual influx of pollution was introduced primarily by rainfall [56], which can dilute pollutants in the rivers but also increase surface runoff and transport soil nutrients to the rivers, where sediment and pollutants are concentrated downstream due to the water flow [21]. The high rainfall amount and frequency in 2013 supplied nutrients to the rivers, making the variation before and after the monsoons relatively small. However, the small rainfall amount and low rainfall frequency in 2015 supplied fewer nutrients to nutrient-rich rivers, creating a larger variation before and after the monsoon.

### 3.4. Relationship between Rainfall Patterns and Environmental Factors

Table 4 presents the results of a correlation analysis between the rainfall pattern and environmental variabilities in the four major water courses. The water temperature variation and rainfall pattern were negatively correlated in all rivers (*p* < 0.05). The DO variation showed a correlation only in the Mangyeonggang River and Sapgyocheon stream (*p* < 0.05 for all), and pH showed a negative correlation only in the Dongjin River (*p* < 0.05 for all). Variation in electrical conductivity was negatively correlated with rainfall amount in the Geumgang River and Dongjingang River before sampling but positively correlated with that of the Mangyeonggang River and Sapgyocheon stream. The Mangyeonggang River and Sapgyocheon stream had many D-class soils with low permeability that exhibited high runoff during rainfall events [13]. Because of this, the rainfall pattern in these two rivers had a positive correlation with the variation in electrical conductivity. In addition, the Sapgyocheon stream, where the proportion of land devoted to urban development was high, exhibited a close correlation between rainfall frequency and electrical conductivity. The variations of NO_3_-N and TP showed a significant correlation with rainfall before the second sampling in the Mangyeonggang River and Sapgyocheon stream. A negative correlation was evident in the Mangyeonggang River, and a positive correlation was revealed in the Sapgyocheon stream. The Sapgyocheon stream had a high proportion of urban areas with high pollution levels and rich nutrients. Urban areas have large areas of impermeable surfaces and high rainwater runoff during rainfalls, which supplies fine sediment and pollutants to the rivers [57,58]. This explains the higher correlation between the rainfall amount in the Sapgyocheon stream before the second sampling and the variation in nutrients.

### 3.5. Epilithic Diatom Communities

The epilithic diatoms that appeared for three years in the rivers in the mid-western region of the Korean Peninsula represented 222 taxa consisting of 2 orders, 3 suborders, 9 families, 37 genera, 200 species, 19 varieties, 2 forma, and 1 subspecies. The variation of major species in each water system before and after the monsoon is as follows (Figure 7).

Geumgang River: Five major species, including *Melosira varians* C. Agardh, *Navicula minima* Grunow, and *Aulacoseira granulata* (Ehrenberg) Simonsen, were found. The highest-standing crops were *Melosira varians* and *Nitzschia palea* (Kützing) W. Smith before monsoons and *Aulacoseira granulata* and *Melosira varians* after monsoons. The standing crop of *Melosira varians*, which was high throughout the survey period, occurred primarily in eutrophic waters [59]. The standing crop increased more in 2015 than in 2013 and 2014, and *Navicula minima*, *Aulacoseira granulate*, and *Cyclotella meneghiniana* Kützing increased after the monsoons.

Mangyeonggang River: Three major species, *Nitzchia palea*, *Melosira varians*, and *Fragilaria elliptica* Schumann, were found. The highest-standing crops were *Melosira varians* and *Nitzchia palea* before the monsoons and *Nitzchia palea* and *Fragilaria elliptica* after the monsoons. *Nitzchia palea*, which was a high-standing crop throughout the survey period, was an indifferent species. It was resistant to pollution and appeared as a dominant species in polluted waters [60]. The standing crop of *Nitzchia palea* and *Melosira varians* increased in 2013, and the standing crop of *Melosira varians* increased in 2014. In 2015, the populations of *Melosira varians* and *Fragilaria elliptica*, which were floating species, increased significantly.

Dongjingang River: Five major species, including *Nitzchia palea*, *Melosira varians*, and *Nitzchia amphibia* Grunow, were found. The highest-standing crops were *Nitzchia palea* and *Melosira varians* before the monsoons and *Nitzschia palea* and *Nitzschia amphibia* after the monsoons. The standing crop of *Nitzschia palea* increased in 2013, and the most standing crops decreased in 2014. The proportion of *Nitzschia amphibia* significantly increased in 2015. It is known as an indifferent species and often appears in nutrient-rich waters [59].

Sapgyocheon stream: Six major species, including *Navicula minima*, *Nitzchia palea*, and *Gomphonema lagenula* Kützing appeared. The highest-standing crops were *Nitzchia palea* and *Navicula minima* before and after the monsoons. *Nitzchia palea* is known to appear in nutrient-rich waters as a dominant species [60], and *Navicula minima* are known to grow in a wide range from freshwater to weak brackish water. The standing crops of major species decreased in 2013 and 2014, and most standing crops distinctively increased in 2015. Most standing crops of the major species in the Sapgyocheon stream significantly decreased after the monsoons of 2013 and 2014 and increased in 2015. In 2013 and 2014, when the rainfall amount and frequency were relatively large, the standing crop of epilithic diatoms decreased after the monsoons. This was the result of the ease of detachment of epilithic diatoms caused by the flow that occurred during rainfall and erosion by suspended sediment [61]. However, in 2015, when the rainfall amount and frequency were small, the standing crop increased because detachment of epilithic diatoms occurred less frequently, and growth was dominant due to nutrient uptake. In particular, the Sapgyocheon stream, where the pollution level and concentration of nutrients were high, saw a significant increase in the standing crop of epilithic diatoms.

### 3.6. Changes in Indicator Species

To identify epilithic diatoms that can represent variable environments along with rainfall on the rivers in the mid-western region of the Korean Peninsula, ISA was performed on each group’s biological community (Table 5 and Table 6):

Geumgang River: Five indicator species were identified before the monsoon, including *Navicula capitatoradiata* H. Germain ex Gasse and *Achnanthes hungarica* (Grunow) Grunow in 2013, *Navicula veneta* Kützing in 2014, and four species including *Cyclotella stelligera* (Cleve & Grunow) Van Heurck and *Navicula cryptotenella* Lange-Bertalot in 2015 (Table 5). *Navicula capitatoradiata***,** whose indicator value was high in 2013, is an indifferent species and inhabits brackish to freshwater [62]. *Navicula veneta*, which was an indicator species in 2014, was an indifferent species and often found in brackish waters with high electrical conductivity and heavily eutrophic waters [62]. *Navicula crytotenella*, which was an indicator species in 2015, was an indifferent species. It is known to inhabit oligotrophic to eutrophic waters but was rarely distributed in heavily polluted areas [62]. The indicator species in the Geumgang River after the monsoons were *Gomphonema parvulum* (Kützing) Kützing and *Aulacoseira ambigua* (Grunow) Simonsen in 2013, three species, including *Navicula veneta* and *Navicula amphiceropsis* Lange-Bertalot & U. Rumrich in 2014, and *Aulacoseira granulata* var. *angustissima* (O. Müller) Simonsen and *Navicula schroeteri* F. Meister in 2015. *Gomphonema parvulum*, which was dominant in 2013, appeared in eutrophic water bodies where there was considerable sewage inflow [63]. *Navicula veneta*, which was the same indicator species before the monsoon, appeared with a high indicator value after the monsoon in 2014. *Aulacoseira granulata* var. *angustissima*, which was an indifferent species, appeared in 2015. The species that appeared in the Geumgang River before the monsoons showed a high indicator value in indifferent species. Saproxenous species appeared after the monsoon in 2013, when the rainfall amount and frequency were large, and the same species appeared in 2014 before and after the monsoon. Indifferent species appeared after the monsoon in 2015.

Mangyeonggang River: Three indicator species were identified before the monsoon in 2013, including *Fragilaria bidens* Heiberg and *Gomphonema pseudoaugur* Lange-Bertalot and *Navicula viridula* var. *rostellata* (Kützing), Cleve and *Navicula trivialis* Lange-Bertalot were identified in 2014, followed by *Fragilaria capucina* Desmazières, *Cyclotella stelligera*, and *Surirella angusta* Kützing in 2015 (Table 5). *Fragilaria bidens*, *Navicula viridula* var. *rostellata*, and *Fragilaria capucina*, which had the highest indicator value each year, are known to inhabit largely mesotrophic waters and prefer weak alkalinity [46,64]. Three indicator species were identified after the monsoon in the Mangyeonggang River, including *Cymbella affinis* Kützing and *Nitzschia inconspicua* Grunow, five species, including *Cocconeis placentula* var. *lineata* (Ehrenberg) Van Heurck and *Navicula gregaria* Donkin were identified in 2014, followed by *Cymbella leptoceros* (Ehrenberg) Kützing and *Navicula decussis* Østrup in 2015. *Cymbella affinis*, which appeared in 2013, is known to prefer waters with average levels of electrolytes [65]. *Cocconeis placentula* var. *lineata*, which appeared in 2014, is often found in mesotrophic and eutrophic waters [59]. *Cymbella leptoceros*, which appeared in 2015, is known to prefer mesotrophic waters [65]. Species that inhabit mesotrophic waters appeared in the Mangyeonggang River as indicator species before the monsoons. However, species that preferred mesotrophic waters appeared after the monsoons.

Dongjingang River: Eight indicator species appeared before the monsoon, including *Cocconeis placentula* var. *lineata* and *Aulacoseira ambigua* in 2013. Six species, including *Gomphonema parvulum* and *Fragilaria bidens,* appeared in 2014, and *Cyclotella atomus* Hustedt was found in 2015 (Table 5). *Cocconeis placentula* var. *lineata*, *Gomphonema parvulum*, and *Cyclotella atomus*, whose indicator values were high, were indifferent species that frequently appeared in eutrophic rivers [63]. Six species, including *Navicula recens* (Lange-Bertalot) Lange-Bertalot and *Navicula notha* J.H. Wallace, appeared in 2013 as indicator species after the monsoons. An indicator species in 2014, *Fragilaria delicatissima* Proshkina-Lavrenko, did not appear in 2015. *Navicula recens*, which appeared in 2013, is an indifferent species that is widely distributed in eutrophic rivers with high electrical conductivity [62]. *Fragilaria delicatissima*, which appeared in 2014, is known to prefer a neutral pH and is indifferent to organic pollutants [33]. Species that often appeared in eutrophic rivers were identified as indicator species before the monsoons in the Dongjingang River. After the monsoons, species that were often distributed in eutrophic streams with high electrical conductivity appeared in 2013, when the rainfall amount and frequency were large. In 2014, indifferent species appeared, and, no indicator species appeared in 2015.

Sapgyocheon stream: Indicator species before the monsoons were *Navicula cari* Ehrenberg and *Fragilaria bidens* in 2013, *Navicula cryptocephala* Kützing in 2014, and *Fragilaria elliptica* and *Navicula veneta* in 2015 (Table 5). *Navicula cari* and *Fragilaria elliptica*, whose indicator values were high in 2013 and 2015, preferred fresh or eutrophic waters with high electrical conductivity [62,66]. *Navicula cryptocephala*, whose indicator value was high in 2014, was known to appear frequently in waters with high electrical conductivity [62]. As an indicator species after the monsoons, *Navicula seminulum* Grunow and *Nitzschia subacicularis* Hustedt appeared in 2013, preferring eutrophic waters. In 2015, *Nitzschia paleacea* (Grunow) Grunow and *Cyclotella stelligera* appeared. In particular, *Cyclotella stelligera* is known to prefer oligotrophic or mesotrophic waters [67]. In the Sapgyocheon stream before the monsoons, species that preferred eutrophic waters with high conductivity emerged. In 2013, when the rainfall amount and frequency were high after the monsoons, species that preferred eutrophic waters with high pollution were found, whereas species that preferred mesotrophic waters appeared in 2015 when the rainfall amount and frequency were small.

In summary, the ecological characteristics of the indicator species that appeared in the Geumgang River, Mangyeonggang River, and Dongjingang River before and after the monsoon were not distinctively different. In the Sapgyocheon stream, saproxenous species that preferred eutrophic waters appeared before the monsoons each year, and after the monsoons, saproxenous species appeared in 2013. In 2015, when the rainfall amount and frequency were low, species that preferred mesotrophic waters appeared, indicating that the ecological characteristics of the indicator species were distinctively differentiated according to the rainfall pattern.

### 3.7. Community Variation

To identify the distribution of epilithic diatoms before and after the monsoon, PCA was conducted using the number of species that appeared for three years and the standing crop of each species. The analysis revealed that the epilithic diatom community could be divided by survey period. Before the monsoon in 2013, diatoms were distributed in the positive direction along axis 1, with a similar trend evident after the monsoon. The diatoms were distributed in both positive and negative directions before and after the monsoons in 2014. In 2015, they were distributed in the negative region along axis 2 (Figure 8a). To quantitatively identify the effect of rainfall on the epilithic diatom community, a CDI was calculated and then comparatively analyzed by river and year (Figure 8b).

The results of the analysis of the CDI in each river showed that the highest CDI (ca. 5.50) occurred in 2015 when the rainfall amount and frequency were small in all rivers (Figure 8c). The Sapgyocheon stream exhibited the lowest value is 2013 and the highest value in 2015 (ANOVA, *p* < 0.01). Rainfalls cause an influx of nutrients from urban and agricultural areas to the rivers [68,69], and more runoff occurs on impervious surfaces in urban areas or soils with low permeability [13,70]. The high rainfall amount and frequency in 2013 caused an influx of nutrients and pollutants into nutrient-rich rivers, making no significant difference in the environment before and after the monsoon, reducing community variation. However, the low rainfall amount and frequency improved water quality, resulting in an environmental change and large community variation before and after the monsoons. This is consistent with the changes in environmental factors.

In a study by Cho et al. (2020) [37], the variation in the epilithic diatom community was large when the rainfall amount was high, which was the opposite result of the present study. The survey sites of Cho et al. (2020) were the main tributaries of the Imjingang River, which is not a large river compared with those in this study. In contrast, the present study selected survey sites from mid- to downstream reaches of relatively large rivers with multiple tributaries. Matter in upstream reaches is transported downstream, and pollutants that are introduced from tributaries [71], non-point pollution sources, point pollution sources, and surrounding lands are concentrated in downstream areas [72]. Intensive rainfall may dilute nutrients but can also introduce a large number of nutrients to rivers by increasing surface runoff [11,21]. A large amount of rainfall did not create an environmental difference from that before the monsoon due to the influx of various components, whereas a small amount of rainfall introduced a low level of nutrients and pollutants to the rivers, making an environmental difference. When epilithic diatoms move from eutrophic waters to an unpolluted environment with fewer nutrients, their community composition changes [73].

In conclusion, the largest change in the epilithic diatom community occurred in 2015, when the habitat environment changed distinctively due to a small amount of rainfall, and the Sapgyocheon stream experienced significant yearly changes because of the large number of urban impermeable surfaces and low-permeability soils.

### 3.8. Relationship between Rainfall Patterns and Epilithic Diatom Communities

The relationship between community variation and rainfall showed a mainly negative correlation, with a significant correlation seen in the Mangyeonggang River and Sapgyocheon stream (Table 6). Sapgyocheon stream showed a weak correlation between the sum of rainfall one month before sampling and the CDI but a significant negative correlation with entire rainfall during the monsoon periods (*p* < 0.05). The sum of rainfalls for one and two weeks in the second survey and the rainfall amount during the monsoon periods showed a significant correlation with the CDI in the Sapgyocheon stream, and the rainfall amount within two weeks had the highest correlation (*p* < 0.01).

The CDI and rainfall frequency also showed a primarily negative correlation, with a significant correlation seen in the Mangyeonggang River and Sapgyocheon stream. The frequency of rainfall events ≥30 mm and ≥50 mm influenced the CDI in the Mangyeonggang River. The highest correlation was with the intensity and frequency of rainfall ≥30 mm. In the Sapgyocheon stream, the frequency of rainfall ≥10 mm, ≥30 mm, ≥50 mm, and ≥70 mm exhibited a correlation, and the frequency of rainfall ≥10 mm had the highest correlation with the CDI (*p* < 0.01).

The rainfall pattern and the CDI were significantly correlated in the Mangyeonggang River and Sapgyocheon stream, which had a high percentage of soil with low permeability. In particular, the frequency of rainfall ≥10 mm had a high correlation with the biotic change in the Sapgyocheon stream, which had a high level of pollution.

## 4. Conclusions

A total of 222 taxa of diatoms emerged in four rivers in the mid-western region of the Korean Peninsula for 3 years, and six species, including *Nitzschia palea*, were dominant. They were mostly indifferent species or species that were dominant in eutrophic waters.

The proportion of soils with low permeability was higher in the Mangyeonggang River and Sapgyocheon stream than in the Geumgang River and Dongjingang River. The proportion of urban areas was the highest around the Sapgyocheon stream. A small amount of rainfall and low rainfall frequency caused an influx of pollutants and nutrients into the rivers, thereby causing a large environmental change in the Sapgyocheon stream before and after the monsoons.

Particularly, the standing crops of the main species in the Sapgyocheon stream after the monsoons decreased in 2013 and 2014 and increased in 2015, when the rainfall amount and frequency were small. The ecological characteristics of the indicator species before and after the monsoon according to rainfall patterns did not show distinct differences in the Geumgang River, Mangyeonggang River, and Dongjingang River. However, in 2013, when the rainfall amount and frequency were high in the Sapgyocheon stream, species that preferred eutrophic waters appeared, and in 2015, species that preferred mesotrophic waters appeared, differentiating these features by year. The CDI calculation results revealed that the highest value was produced in 2015 when a small amount of rainfall occurred, and this was most evident in the Sapgyocheon stream. A relationship between the rainfall pattern and CDI was identified, with a negative correlation evident overall. In particular, the two-week rainfall amount and frequency of rainfall ≥10 mm had the highest correlation with the CDI in the Sapgyocheon stream.

In conclusion, the distribution of epilithic diatoms inhabiting the four main watercourses in the mid-western region of the Korean Peninsula was significantly affected by rainfall amount and frequency (intensity) in the monsoon period. In addition, the epilithic diatom community variation was also closely correlated with land use and soil characteristics around the surveyed rivers. Finally, the community variation was larger as the rainfall amount and frequency (intensity) were lower in the Sapgyocheon stream, which exhibited low permeability and high nutrient concentrations.

## Figures and Tables

**Figure 1 ijerph-20-04099-f001:**
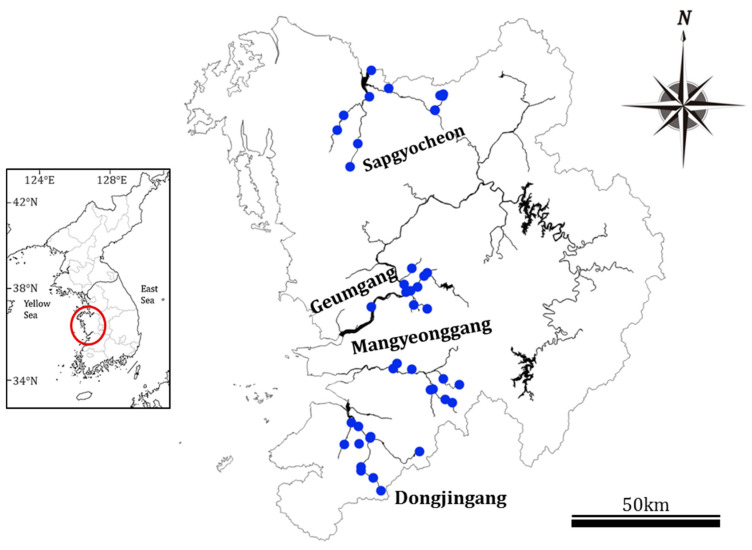
Location of the 42 sampling sites in four central western major streams of the Korean Peninsula. Aquatic environmental factors and diatoms were sampled twice a year from 2013 to 2015 (Geumgang; Geumgang River, Mangyeonggang; Mangyeonggang River, Dongjingang; Dongjingang River, Sapgyocheon; Sapgyocheon stream).

**Figure 2 ijerph-20-04099-f002:**
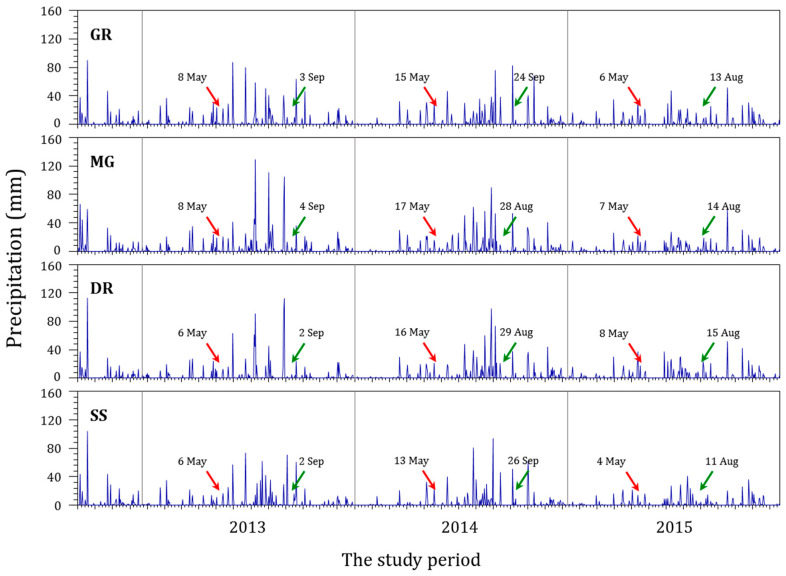
Daily precipitation for the four central western major streams of the Korean Peninsula between 2013 and 2015 (GR: Geumgang River, MR: Mangyeonggang River, DR: Dongjingang River, SS: Sapgyocheon stream). The arrows are sampling dates before (red) and after (green) the monsoon.

**Figure 3 ijerph-20-04099-f003:**
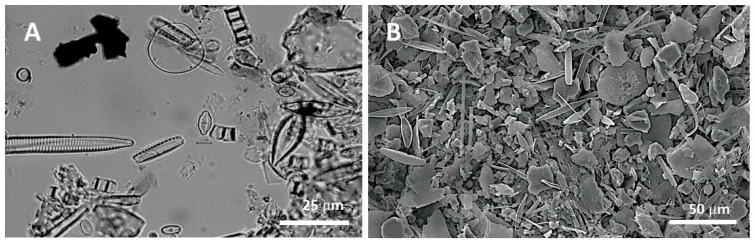
Light (**A**) and scanning electron microscopic photos (**B**) were prepared for the species identification and enumeration of epilithic diatoms collected from the sampling site.

**Figure 4 ijerph-20-04099-f004:**
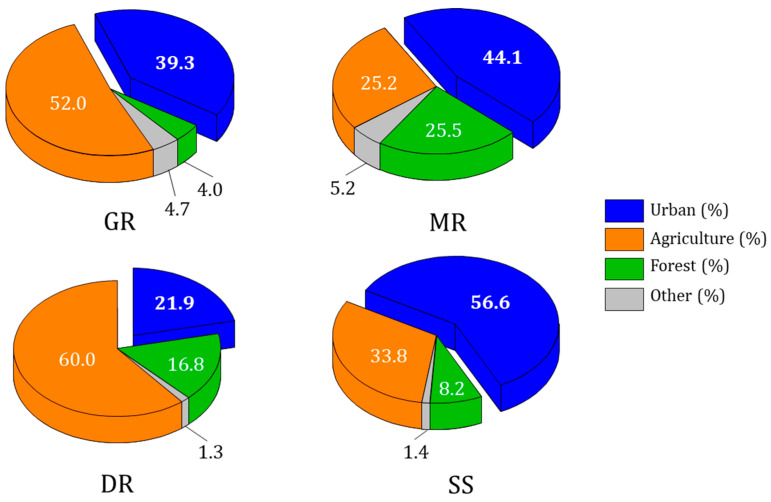
Land use of four central western major streams of the Korean Peninsula (GR: Geumgang River, MR: Mangyeonggang River, DR: Dongjingang River, SS: Sapgyocheon stream).

**Figure 5 ijerph-20-04099-f005:**
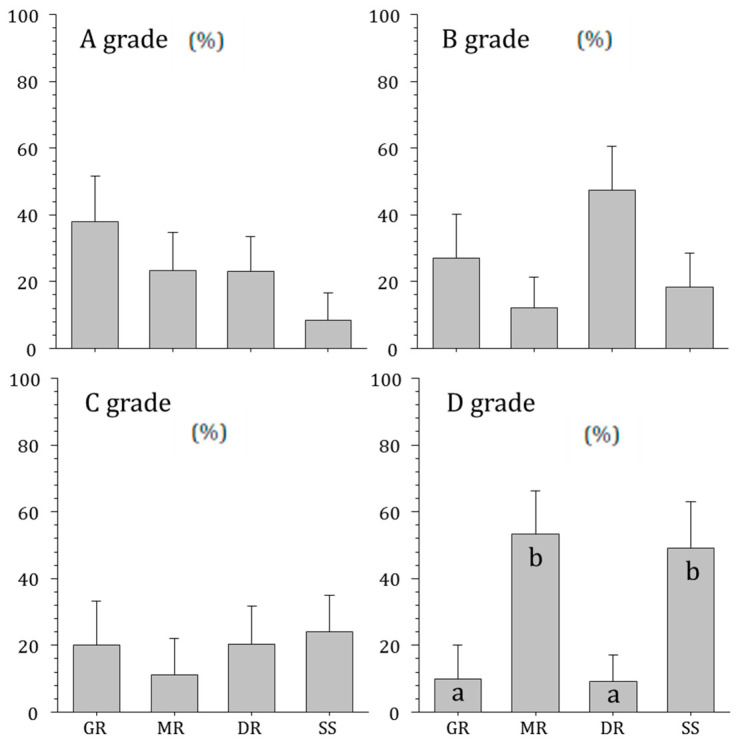
Soil characteristics of four central western major streams of the Korean Peninsula (GR: Geumgang River, MR: Mangyeonggang River, DR: Dongjingang River, SS: Sapgyocheon stream). Small alphabets (a, b) was Tukey’s post hoc test.

**Figure 6 ijerph-20-04099-f006:**
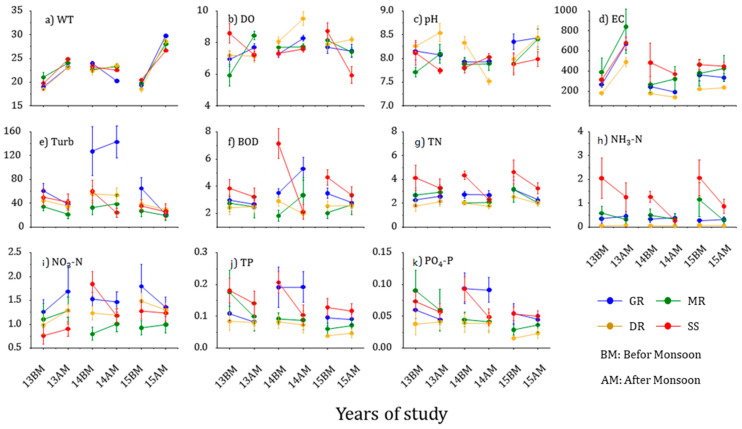
Temporal variability of the physical-chemical variables in the four central western major streams of the Korean Peninsula from 2013 to 2015 (BM: before monsoon, AM: after the monsoon, GR: Geumgang River, MR: Mangyeonggang River, DR: Dongjingang River, SS: Sapgyocheon stream). WT: water temperature (°C), DO: dissolve oxygen (mg/L), BOD: biochemical oxygen demand (mg/L), EC: electronic conductivity (µS/Cm), Turb.: Turbidity (NTU). TN: total nitrogen (mg/L), TP: total phosphorus (mg/L), ${\mathrm{NH}}_{3}$-N, ${\mathrm{NO}}_{3}$-N, ${\mathrm{PO}}_{4}$-P: mg/L.

**Figure 7 ijerph-20-04099-f007:**
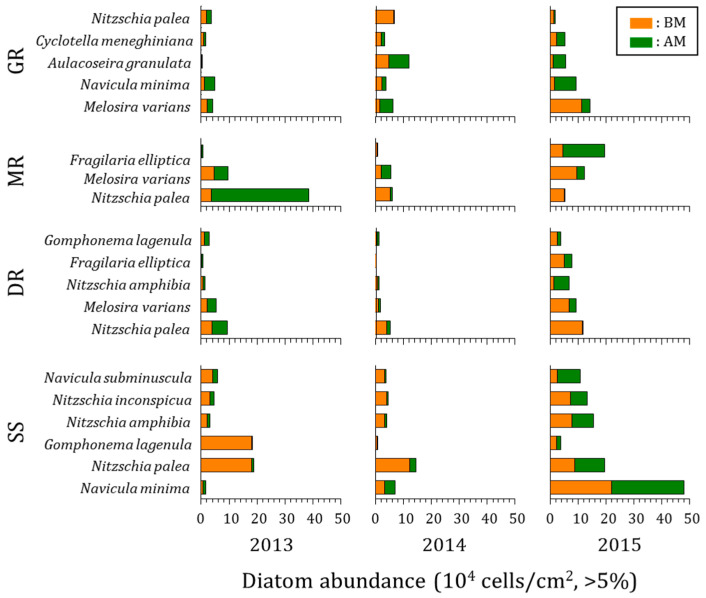
Epilithic diatom abundance of the most abundant species in four central western major streams of the Korean Peninsula before and after monsoons from 2013 to 2015. Top 5% of total abundance (GR: Geumgang River, MR: Mangyeonggang River, DR: Dongjingang River, SS: Sapgyocheon stream), BM: before monsoon, AM: after the monsoon.

**Figure 8 ijerph-20-04099-f008:**
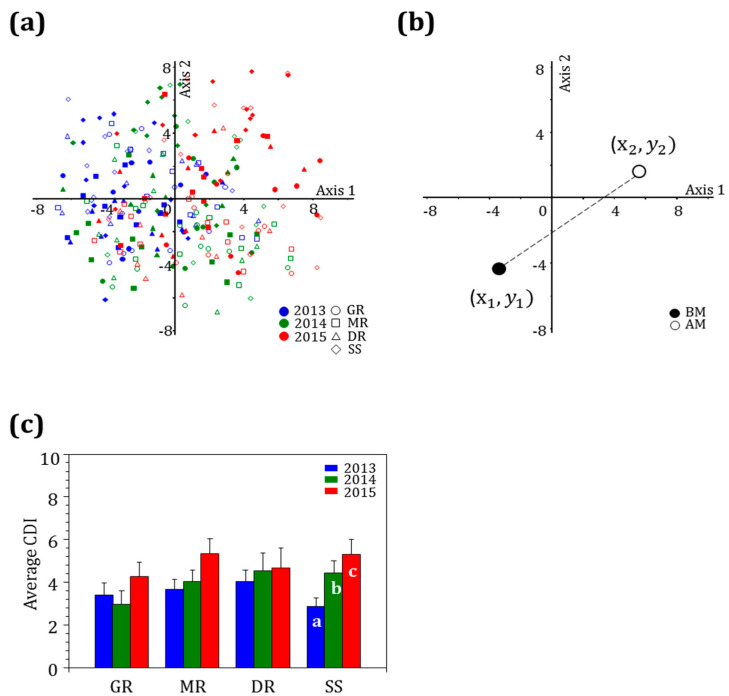
Epilithic diatom community characteristics before and after the monsoon in four central western major streams (2013–2015). (**a**) Principal component analysis (PCA) ordination, (**b**) Community dynamic index (CDI) calculation, (**c**) an average of the dynamic community index of epilithic diatom (GR: Geumgang River, MR: Mangyeonggang River, DR: Dongjingang River, SS: Sapgyocheon stream). Superscripts ^a, b,^ and ^c^ are Tukey’s post hoc test after the Bonferroni test.

**Table 1 ijerph-20-04099-t001:** Characteristics of soil in four central western major streams of the Korean Peninsula between 2013 and 2015.

Characters	Characteristics of Soil			
Drainage Class	Very Good	Good	Bad	Very Bad
Permeability (cm/h)	Very fast (>12.0)	Fast (12.0–6.0)	Slow (6.0–5.0)	Very slow (<0.5)
Hydrological soil group (grade)	A	B	C	D

**Table 2 ijerph-20-04099-t002:** Characteristics of soil in four central western major streams of the Korean Peninsula between 2013 and 2015 (GR: Geumgang River, MR: Mangyeonggang River, DR: Dongjingang River, SS: Sapgyocheon stream). The variable wk: week, Example: 1 wk, means that total precipitation for a day before the second sampling date through the study period.

Sites		Sum of Precipitation for a Certain Period before the Second Sampling (mm)					Precipitation Frequency in Monsoon Period (No.)				
		1 wk	2 wk	3 wk	4 wk	Total	≥10 mm	≥30 mm	≥50 mm	≥70 mm	≥90 mm
GR	2013	10.0	81.9	81.9	86.6	631.5	17.0	6.0	4.0	2.0	0.0
	2014	77.0	77.0	81.1	127.6	622.0	15.9	6.9	1.9	1.9	0.0
	2015	17.6	17.6	46.5	63.7	273.3	10.0	1.0	0.0	0.0	0.0
MR	2013	14.6	191.4	194.5	194.5	853.3	18.2	8.0	4.0	3.0	3.0
	2014	33.7	89.6	103.6	154.3	752.4	22.1	7.1	5.1	1.0	0.0
	2015	11.7	11.7	25.4	64.0	270.5	10.0	0.0	0.0	0.0	0.0
DR	2013	0.3	208.5	228.5	228.5	726.0	15.3	7.1	4.9	3.0	3.0
	2014	97.9	264.2	302.4	398.7	667.6	19.3	6.1	3.2	1.9	0.9
	2015	18.7	26.2	39.1	77.0	326.6	11.0	2.0	0.0	0.0	0.0
SS	2013	70.7	100.1	103.1	121.8	622.6	15.9	7.9	4.1	2.1	0.3
	2014	55.5	55.5	55.5	109.5	913.2	13.0	6.0	3.0	2.0	1.0
	2015	6.5	46.5	115.6	120.6	272.1	9.2	1.1	0.0	0.0	0.0

**Table 3 ijerph-20-04099-t003:** Physical-chemical factors of the epilithic diatom communities in four central western major streams of the Korean Peninsula from 2013 to 2015 (GR: Geumgang River, MR: Mangyeonggang River, DR: Dongjingang River, SS: Sapgyocheon stream).

Variables		GR	MG	DG	SS	F	*p*
WT	BM	20.8 ± 0.5	21.2 ± 0.5	19.9 ± 0.5	21.2 ± 0.4	1.697	0.171
	AM	24.4 ± 0.8 **	25.1 ± 0.6 **	25.1 ± 0.5 **	24.7 ± 0.4 **	0.351	0.788
DO	BM	7.3 ± 0.2 ^a^	7.3 ± 0.3 ^a^	7.7 ± 0.2 ^a,b^	8.2 ± 0.3 ^b^	2.916	0.037
	AM	7.8 ± 0.2 ^b^	7.9 ± 0.2 ^b^	8.3 ± 0.3 ^b^	6.9 ± 0.3 ^a(^**^)^	6.880	<0.001
BOD	BM	3.3 ± 0.2 ^b^	2.2 ± 0.3 ^a^	2.6 ± 0.2 ^a,b^	5.2 ± 0.5 ^c^	15.037	<0.001
	AM	3.6 ± 0.4	2.8 ± 0.5	2.4 ± 0.2	2.9 ± 0.4 ^(^**^)^	1.796	0.152
pH	BM	8.1 ± 0.1 ^b^	7.8 ± 0.1 ^a^	8.2 ± 0.1 ^b^	7.9 ± 0.1 ^a^	2.899	0.038
	AM	8.1 ± 0.1	8.1 ± 0.1 *	8.2 ± 0.1	7.9 ± 0.1	1.497	0.219
EC	BM	287.9 ± 23.1 ^a,b^	342.4 ± 67.6 ^b^	191.3 ± 10.1 ^a^	420.1 ± 67.2 ^b^	4.058	0.009
	AM	397.0 ± 44.1 ^a,b,^*	530.0 ± 112.6 ^b^	288.2 ± 30.7 ^a,^**	498.7 ± 35.7 ^b^	3.499	0.018
Turb	BM	84.2 ± 16.0 ^b^	31.2 ± 5.4 ^a^	46.4 ± 6.0 ^a^	48.5 ± 8.6 ^a^	4.787	0.003
	AM	67.5 ± 13.6 ^b^	26.5 ± 6.4 ^a^	39.2 ± 5.7 ^a^	30.7 ± 6.7 ^a^	4.441	0.005
TN	BM	2.72 ± 0.33 ^a^	2.63 ± 0.4 ^a^	2.12 ± 0.19 ^a^	4.37 ± 0.49 ^b^	7.323	<0.001
	AM	2.51 ± 0.26	2.35 ± 0.26	1.97 ± 0.17	2.95 ± 0.31 ^(^*^)^	2.670	0.051
NH_3_-N	BM	0.32 ± 0.08 ^a^	0.75 ± 0.26 ^a^	0.06 ± 0.01 ^a^	1.79 ± 0.38 ^b^	10.193	<0.001
	AM	0.39 ± 0.09 ^a,b^	0.31 ± 0.10 ^a^	0.07 ± 0.01 ^a^	0.80 ± 0.23 ^b(^*^)^	4.700	0.004
NO_3_-N	BM	1.53 ± 0.18 ^b^	0.94 ± 0.13 ^a^	1.23 ± 0.1 ^a,b^	1.29 ± 0.14 ^a,b^	2.689	0.049
	AM	1.51 ± 0.20	1.10 ± 0.12	1.26 ± 0.08	1.10 ± 0.09	2.196	0.092
TP	BM	0.13 ± 0.02 ^b,c^	0.11 ± 0.03 ^a,b^	0.07 ± 0.01 ^a^	0.17 ± 0.02 ^c^	4.798	0.003
	AM	0.12 ± 0.02	0.09 ± 0.02	0.07 ± 0.01	0.12 ± 0.02	2.563	0.058
PO_4_-P	BM	0.07 ± 0.01 ^b^	0.05 ± 0.01 ^a,b^	0.03 ± 0.01 ^a^	0.07 ± 0.01 ^b^	4.076	0.008
	AM	0.06 ± 0.01	0.05 ± 0.01	0.03 ± 0.01	0.05 ± 0.01	1.628	0.186

WT: water temperature (°C), DO: dissolve oxygen (mg/L), BOD: biochemical oxygen demand (mg/L), EC: electronic conductivity (μS/Cm), Turb.: Turbidity (NTU). TN: total nitrogen (mg/L), TP: total phosphorus (mg/L), ${\mathrm{NH}}_{3}$-N, ${\mathrm{NO}}_{3}$-N, ${\mathrm{PO}}_{4}$-P: mg/L, BM: Before Monsoon, AM: After Monsoon, Small alphabets (a, b and c) were Tukey’s post hoc test, Asterisk was paired *t*-test for BM and AM sampling (*: *p* < 0.05, **: *p* < 0.01), * and (*): significantly increase and decrease.

**Table 4 ijerph-20-04099-t004:** Correlation coefficients between precipitation pattern and temporal variabilities of environmental factors in four central western major streams of the Korean Peninsula from 2013 to 2015 (GR: Geumgang River, MR: Mangyeonggang River, DR: Dongjingang River, SS: Sapgyocheon stream). For example, 1wk means the total precipitation for a day before the second sampling date through the study period.

Variables		Sum of Precipitation (mm)					Precipitation Frequency in Monsoon Period (No.)				
		1 wk	2 wk	3 wk	4 wk	Total	≥10 mm	≥30 mm	≥50 mm	≥70 mm	≥90 mm
WT	GR	−0.756 **	−0.515 **	−0.636 **	−0.878 **	−0.659 **	−0.573 **	−0.712 **	−0.382 *	−0.624 **	−
	MR	−	−	−	−	−0.539 **	−0.610 **	−0.557 **	−0.592 **	−	−
	DR	−0.393 *	−0.703 **	−0.713 **	−0.786 **	−0.610 **	−0.780 **	−0.586 **	−0.536 **	−0.551 **	−
	SS	−0.384 *	−	0.760 **	0.748 **	−0.432 **		−0.340 *	−0.357 *	−0.432 **	−0.614 **
DO	GR	−	−	−	−	−	−	0.376*	−	−	−
	MR	−	0.509 **	0.485 *	0.394 *	0.519 **	−	0.491 **	0.421 *	0.645 **	0.643 **
	DR	−	−	−	−	−	−	−	−	−	−
	SS	0.495 **	−	−0.580 **	−0.516 **	0.522 **	0.375 *	0.474 **	0.484 **	0.531 **	0.537 **
pH	GR	−	−	−	−	−	−	−	−	−	−
	MR	−	−	−	−	−	−	−	−	−	−
	DR	−0.557 **	−0.560 **	−0.579 **	−0.684 **	−0.442 *	−0.717 **	−0.424 *	−0.364 *	−0.358 *	−
	SS	−	−	−	−	−	−	−	−	−	−
EC	GR	−0.769 **	−	−	−0.516 **	−	−	−	−	−	−
	MR	−	0.438 *	0.424 *	−	−	−	−	−	0.494 **	0.521 **
	DR	−0.820 **	−	−	−	−	−	−	−	−	0.475 **
	SS	−	0.483 **	−	−	0.295	0.420 *	0.352 *	0.337 *	−	−
TP	GR	−	−	−	−	−	−	−	−	−	−
	MR	−	−0.445 *	−0.435 *	−0.385 *	−	−	−	−	−0.436*	−0.455*
	DR	−	−	−	−	−	−	−	−	−	−
	SS	−	−	0.425 **	0.395 *	−	−	−	−	−0.332 *	−0.391 *
PO_4_-P	GR	−	−	−	−	−	−	−	−	−	−
	MR	−	−	−	−	−	−	−	−	−0.385 *	−
	DR	−	−	−	−	−	−	−	−	−	−
	SS	−	−	0.389 *	0.362 *	−	−	−	−	−	−0.333 *

WT: water temperature (°C), DO: dissolve oxygen (mg/L), BOD: biochemical oxygen demand (mg/L), EC: electronic conductivity (μS/Cm), Turb.: Turbidity (NTU). TN: total nitrogen (mg/L), TP: total phosphorus (mg/L), ${\mathrm{NH}}_{3}$-N, ${\mathrm{NO}}_{3}$-N, ${\mathrm{PO}}_{4}$-P: mg/L, wk: week, Asterisk was a correlation for environmental factors and Rain patterns (*: *p* < 0.05, **: *p* < 0.01).

**Table 5 ijerph-20-04099-t005:** Indicator species in Geumgang river (GR) and Mangyeonggang river (MR), Dongjingang River (DR), and Sapgyocheon stream (SS) of the Korean Peninsula from 2013 to 2015.

		Species	2013	2014	2015	*p*
GR	BM	*Navicula capitatoradiata*	61	1	4	0.003
		*Achnanthes hungarica*	58	1	1	0.002
		*Navicula contenta*	42	2	0	0.008
		*Aulacoseira ambigua*	40	0	0	0.016
		*Cocconeis placentula* var. *lineata*	40	0	0	0.022
		*Navicula veneta*	0	40	0	0.020
		*Cyclotella stelligera*	0	0	70	0.001
		*Navicula cryptotenella*	0	0	70	0.001
		*Gomphonema lagenula*	7	0	69	0.001
		*Fragilaria delicatissima*	0	5	45	0.004
	AM	*Gomphonema parvulum*	53	1	0	0.001
		*Aulacoseira ambigua*	50	0	0	0.008
		*Navicula veneta*	0	72	1	0.001
		*Navicula amphiceropsis*	0	60	0	0.001
		*Nitzschia levidensis*	0	40	0	0.027
		*Aulacoseira granulata* var. *angustissima*	0	0	40	0.024
		*Navicula schroeteri*	4	1	34	0.021
MR	BM	*Fragilaria bidens 8*	48	21	0	0.009
		*Gomphonema pseudoaugur*	40	0	6	0.032
		*Ulnaria ulna*	10	14	53	0.007
		*Navicula viridula* var. *rostellata*	1	43	5	0.037
		*Navicula trivialis*	2	38	2	0.044
		*Fragilaria capucina*	0	1	81	0.001
		*Cyclotella stelligera*	0	0	44	0.013
		*Surirella angusta*	0	0	44	0.021
	AM	*Cymbella affinis*	48	12	2	0.021
		*Nitzschia inconspicua*	45	14	10	0.031
		*Navicula notha*	44	0	0	0.027
		*Cocconeis placentula* var. *lineata*	0	56	0	0.009
		*Navicula gregaria*	2	48	0	0.007
		*Navicula amphiceropsis*	0	48	2	0.011
		*Cymbella cistula*	2	43	0	0.014
		*Aulacoseira granulata*	0	36	2	0.030
		*Cymbella leptoceros*	0	0	44	0.026
		*Navicula decussis*	0	0	44	0.025
DR	BM	*Cocconeis placentula* var. *lineata*	64	0	0	0.002
		*Aulacoseira ambigua*	55	0	0	0.003
		*Cyclotella pseudostelligera*	55	0	0	0.003
		*Gomphonema angustatum*	55	0	0	0.003
		*Navicula gregaria*	50	2	10	0.006
		*Nitzschia frustulum*	36	0	0	0.024
		*Nitzschia gracilis*	36	0	0	0.030
		*Nitzschia subacicularis*	36	0	0	0.028
		*Gomphonema parvulum*	1	75	1	0.001
		*Fragilaria bidens*	4	65	0	0.002
		*Cyclotella stelligera*	0	63	3	0.001
		*Aulacoseira granulata*	1	52	1	0.005
		*Achnanthes delicatula*	0	36	2	0.022
		*Gomphonema clevei*	1	33	1	0.038
		*Cyclotella atomus*	0	0	45	0.008
	AM	*Navicula recens*	55	0	0	0.002
		*Navicula notha*	38	1	0	0.015
		*Aulacoseira ambigua*	36	0	0	0.026
		*Cyclotella atomus*	36	0	0	0.035
		*Ulnaria acus* (Kützing) Aboal	36	0	0	0.024
		*Cymbella cistula*	26	5	0	0.046
		*Fragilaria delicatissima*	0	45	0	0.003
SS	BM	*Navicula cari*	50	1	0	0.002
		*Fragilaria bidens*	46	3	0	0.001
		*Navicula cryptocephala*	2	48	8	0.005
		*Fragilaria elliptica*	1	1	61	0.001
		*Navicula veneta*	0	4	39	0.005
	AM	*Navicula seminulum*	58	0	0	0.001
		*Nitzschia subacicularis*	36	1	0	0.005
		*Nitzschia paleacea*	0	0	42	0.011
		*Cyclotella stelligera*	0	5	24	0.046

BM: before monsoon, AM: after the monsoon.

**Table 6 ijerph-20-04099-t006:** The correlation coefficients between precipitation pattern and Community Dynamic Index (CDI) in four central western major streams of the Korean Peninsula from 2013 to 2015 (GR: Geumgang River, MR: Mangyeonggang River, DR: Dongjingang River, SS: Sapgyocheon stream).

Sum of Precipitation (mm)	GR	MR	DR	SS	Precipitation Intensity and Frequency (No.)	GR	MR	DR	SS
1 wk	−0.226	−0.101	0.039	−0.377 *	≥10 mm	−0.242	−0.358	0.012	−0.450 **
2 wk	−0.282	−0.265	−0.015	−0.480 **	≥30 mm	−0.266	−0.409 *	−0.066	−0.425 **
3 wk	−0.312	−0.234	−0.033	−0.047	≥50 mm	−0.159	−0.382 *	−0.082	−0.411 *
4 wk	−0.335	−0.173	−0.002	−0.153	≥70 mm	−0.224	−0.351	−0.112	−0.350 *
Total	−0.301	−0.385 *	−0.026	−0.353 *	≥90 mm	0	−0.265	−0.123	0.026

The variable wk: week, Example: 1 wk, means that total precipitation for a day before the second sampling date through the study period. Asterisk was a correlation between environmental factors and Rain patterns (*: *p* < 0.05, **: *p* < 0.01).

## Data Availability

Not applicable.

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
