# Peer review of "The Relationship between Rainfall Pattern and Epilithic Diatoms in Four Streams of Central-Western Korea for Three Years (2013–2015)"

_ijerph, 2023, doi:10.3390/ijerph20054099_

Round 1

Reviewer 1 Report

This manuscript is devoted to an important issue - elucidation of changes in the diversity of diatoms depending on water level, and its results may not only be of regional significance. For example, for other regions in which the level of rivers may rise due to the melting of glaciers it is also important.

However, the manuscript needs significant revision. 1. I recommend writing separate chapters Results and Discussion. 2. The order of placement of tables and figures: I did not find table 6; a figure or a table should be placed on the page immediately after it is mentioned in the text or on the next page from the top. 3. At the first mention of species, the authors of the taxon should be indicated. It is convenient to do this in a table 5. It is possible to indicate the authors of taxa in it, but since the species of diatoms are mentioned earlier in the text (before the reference to Table 5), it should be "raised", that is, inserted into the text earlier. 4. Abbreviations named reservoirs GR; MR, etc. - it is better to introduce in the methods, Highlight a separate paragraph - "Abbreviations", so that the reader can easily find them. Then it will not be necessary to repeat their full names under each figure or in a table. 5. Do not split small tables into 2 pages. Table 4. 6. The caption to Table 4 does not indicate what * and ** stand for. 7. Figure captions should also not be broken - see Figure 5 caption. 8. I can't figure out why there is a "In summary" at the end of page 12? 9. Synedra acus is now a different name. Please check species names and their authors in AlgaeBase.

10. Funding: The project number should be indicated.

Author Response

Reviewer 1

  1. I recommend writing separate chapters Results and Discussion.

Response: Thank for your kind comments, at this point, your comment will be carefully considered in future study. In fact, the authors commonly decided that the flow of our research should be written in narrative form without distinguishing between results and considerations. The main content of this study is to reveal the effects of various rainfall patterns in close proximity, and to highlight the differences through self-comparison between research sites rather than comparing with other previous studies. I hope you understand this point.”

  1. The order of placement of tables and figures: I did not find table 6; a figure or a table should be placed on the page immediately after it is mentioned in the text or on the next page from the top.

Response: Thank for your comment, we revised it.

  1. At the first mention of species, the authors of the taxon should be indicated. It is convenient to do this in a table 5. It is possible to indicate the authors of taxa in it, but since the species of diatoms are mentioned earlier in the text (before the reference to Table 5), it should be "raised", that is, inserted into the text earlier.

Response: Thank for your comment, we mentioned the name of the taxonomy author to the text.

  1. Abbreviations named reservoirs GR; MR, etc. - it is better to introduce in the methods, Highlight a separate paragraph - "Abbreviations", so that the reader can easily find them. Then it will not be necessary to repeat their full names under each figureor in a table. 

Response: Thank for your comments. Although adding a new abbreviation is not in the regulations of this magazine, it is cumbersome, but it is judged that it is common to put an abbreviation in each graph or table. your comment will be carefully considered in future study.

  1. Do not split small tables into 2 pages. Table 4. 

Response: Thank for your comment, we revised it.

  1. The caption to Table 4 does not indicate what * and ** stand for. 

Response: Thank for your comment, we added caption.

  1. Figure captions should also not be broken - see Figure 5 caption. 

Response: Thank for your comment, we revised it.

  1. I can't figure out why there is a "In summary" at the end of page 12? 

Response: Thank for your comment, we deleted “In summary” and added a discussion on Sapgyocheon.

  1. Synedra acusis now a different name. Please check species names and their authors in AlgaeBase.

Response: Thank for your comment, we modified the species name by referring to the AlgaeBase.

Reviewer 2 Report

I have outlined below my understanding and thoughts on it, in the hopes, they will help in further improving its scientific contribution
(1) In the abstract:

-          Add more quantitative data.

(2) In the introduction:

-          Studies on the effects of monsoons on streams in the Korean Peninsula have been 75 conducted on water quality and fish [4,38-40] –( give some explanation for these studies)

 (3): In the results:

-          Need to compare the results with more recent studies from the literature  

Author Response

Reviewer 2

(1) In the abstract: Add more quantitative data.

Response: Thank for your comment, we revised it as followed;

To study the effect of rainfall patterns on diatom communities in four major central western streams on the Korean Peninsula during the monsoon seasons of 2013 through 2015, we measured precipitation, environmental factors, and epilithic diatoms at 42 sites before (May) and after (August and September) each monsoon. The Mangyeonggang river and Sapgyocheon stream (SS) had a high percentage of low-permeability soil, and the stream had the highest proportion (49.1%) of sur-rounding land in urban areas. Precipitation and precipitation frequency was closely correlated with electrical conductivity and nutrients, and this was particularly evident in Sapgyocheon stream. Epilithic diatom abundance for the most abundant species as Navicula minima decreased in the stream 2013 and 2014 and increased in 2015, when precipitation and precipitation frequency were low. This was not clearly distinguishable in the ecological characteristics of each watercourse’s indicator species, except in Sapgyocheon stream. The community dynamic index was highest in 2015 (ca. 5.50) and the annual changes in the index were clearly shown in SS. The precipitation pattern and the community dynamic index were negatively correlated (r=-0.026~-0.385), and the precipitation within 2 weeks (r=-0.480 for SS) before the second sampling and the frequency of 10 mm of precipitation were closely correlated in the stream (r=-0.450 for SS). The distribution of epilithic diatoms in the four watercourses is therefore affected by monsoon precipitation and precipitation frequency, and the community dynamic index is determined by soil characteristics and land use.

(2) In the introduction: Studies on the effects of monsoons on streams in the Korean Peninsula have been conducted on water quality and fish [4, 38-40] –( give some explanation for these studies)

Response: Thank for your comment, we added an explanation to the text.

(3): In the results: Need to compare the results with more recent studies from the literature

Response: Thank for your comment, your comment will be carefully considered in future study. The main content of this study is to reveal the effects of various rainfall patterns in close proximity, and to highlight the differences through self-comparison between research sites rather than comparing with other previous studies. I hope you understand this point.

Reviewer 3 Report

In this paper, the effect of rainfall patterns on diatom communities in four major central western streams during the monsoon seasons of 2013 through 2015 was studied. I have several suggestions as follows before the paper can be accepted.

Introduction:

The authors should review some papers regarding the effect of rainfall patterns on diatom communities. Then, the readers can better understand the status of the current studies. Also, the paper is like a case study, but even for case study, the results still can show some new insight or findings in compared with previous studies.

2.3. Environmental factors

The land use includes a radius of 50 m from sampling place. How do you decide the values? Is there any reference? In addition, do you consider the distance effect, i.e., if the urban is 30 m or 50 m from the sampling place? They should have difference influences.    

2.4. Epilithic diatoms:

The authors may show some in-situ photos for Epilithic diatoms, which help reader to easily understand.

2.6. Community dynamic index

Show the formula used to calculate community dynamic index.

2.7. Data analysis

In Table 2, for the Precipitation frequency in monsoon period (No.), why are the values not integers? As I understand, they should be integers. In addition, how do you decide the threshold “10mm, 30mm, 50mm…”? Is there any reference?

3.2. Land use and soil characteristics

What is y-axis in Figure 4? It is better to provide an example (map, a radius of 50 m from sampling place) of land use.

3.3. Environmental characteristics

In Table 3, please explain what is “BM” and “AM”?

Grammar errors

Line 19: in in

Line 149: 5 × 5 cm2

Line 164: The higher the IndVal (0–100), the greater its indicative power

Line 202: which was a similar

Author Response

Reviewer 3

Introduction: The authors should review some papers regarding the effect of rainfall patterns on diatom communities. Then, the readers can better understand the status of the current studies. Also, the paper is like a case study, but even for case study, the results still can show some new insight or findings in compared with previous studies.

Response: Thank for your comment, we inserted current studies in text.

2.3. Environmental factors: The land use includes a radius of 50 m from sampling place. How do you decide the values? Is there any reference? In addition, do you consider the distance effect, i.e., if the urban is 30 m or 50 m from the sampling place? They should have difference influences.

Response: Thank for your comment, According to the guidelines for Stream/River Ecosystem Survey and Health Assessment by the Ministry of Environment (MOE) of the Republic of Korea, land use within a radius of 50m is investigated when surveying diatoms. We decided based on this.

2.4. Epilithic diatoms: The authors may show some in-situ photos for Epilithic diatoms, which help reader to easily understand.

Response: Thank for your comment, your comment will be carefully considered in future study.

2.6. Community dynamic index: Show the formula used to calculate community dynamic index.

Response: Thank for your comment, we inserted the formula.

2.7. Data analysis: In Table 2, for the Precipitation frequency in monsoon period (No.), why are the values not integers? As I understand, they should be integers. In addition, how do you decide the threshold “10mm, 30mm, 50mm…”? Is there any reference?

Response: Thank for your comment, there are 42 sampling sites (GR 10 sites, MR 9 sites, DR 10 sites, SS 12sites), and the rainfall affected by each site varies slightly. Since the average precipitation value affecting each site was obtained, it does not appear as an integer. In addition, the threshold was decided based on the maximum and minimum values of precipitation from 2013 to 2015.

3.2. Land use and soil characteristics: What is y-axis in Figure 4? It is better to provide an example (map, a radius of 50 m from sampling place) of land use.

Response: Thank for your comment, the y-axis in Figure 4 shows the % for each group soil class, and we added captions to the text.

3.3. Environmental characteristics: In Table 3, please explain what is “BM” and “AM”?

Response: Thank for your comment, we added caption. 

Grammar errors

Line 19: in in

Response: Thank for your comment, we fixed typing error.

Line 149: 5 × 5 cm2

Response: Thank for your comment, we fixed typing error.

Line 164: The higher the IndVal (0–100), the greater its indicative power

Response: Thank for your comment, we deleted the sentence and added a new one.

Line 202: which was a similar

Response: Thank for your comment, we fixed typing error.

Reviewer 4 Report

Dear Authors!

Find attached my suggestions for your manuscript.

line 43-45: refer to them as a running water, stream and river are different terminology.

line 46-53: it is a material and methods part, not an introduction.

line 67: please change it to physical-chemical.

line 148: iron brush? a normal plastic brush maybe a better solution not to break apart a lot of diatoms.

table 2: wk represents what?

figure 4: numbers on the y refers to what? and a and b stands for?

line 246: relatively high? for me it seems ways smaller than the previously mentioned 2 rivers EC

line 249: how do the authors proves that turbidity decreased before the sampling? it is a little controversial… and what do the authors thinks about that decreased turbidity? in normally they need to increase because of the increased velocity, etc…

table 3: physical-chemical change it, need to add units too and meaning of abbreviations.

figure 5: it is quite challenging to see true differences on graphs with different scales. In that form please do not consider showing the next to each other, as reminds

figure 7: BM and AM stand for what?

The monsoon really was the clue which have an effect on the diatom community variation, or just cases changes in environmental facts that can directly affecting the community structure?

Author Response

Reviewer 4

line 43-45: refer to them as a running water, stream and river are different terminology.

Response: Thank for your comment, we revised the text by referring to the meaning of the terminology.

line 46-53: it is a material and methods part, not an introduction.

Response: Thank for your comment, we moved line 45-55 to material and methods (2.1. survey sites).

line 67: please change it to physical-chemical.

Response: Thank for your comment, we changed the word.

line 148: iron brush? a normal plastic brush maybe a better solution not to break apart a lot of diatoms.

Response: Thank for your comment, we used a soft brush to scrub epilithic diatoms.

table 2: wk represents what?

Response: Thank for your comment, we added caption.

figure 4: numbers on the y refers to what? and a and b stands for?

Response: Thank for your comment, we added caption.

line 246: relatively high? for me it seems ways smaller than the previously mentioned 2 rivers EC

Response: Thank for your comment, we fixed typing error.

line 249: how do the authors proves that turbidity decreased before the sampling? it is a little controversial… and what do the authors thinks about that decreased turbidity? in normally they need to increase because of the increased velocity, etc…

Response: Thank for your comment, we corrected the sentence because it was judged that there was a semantic error in line 249. Turbidity fell after monsoon in all groups. Generally, for several days after a large amount of rainfall (approximately 2-4 days), turbidity increases due to high flow rates, but turbidity decreases sharply after several days. The decrease in turbidity is thought to be due to the fact that the second survey was conducted after the water area stabilized after monsoon.

table 3: physical-chemical change it, need to add units too and meaning of abbreviations.

Response: Thank for your comment, we changed the word.

figure 5: it is quite challenging to see true differences on graphs with different scales. In that form please do not consider showing the next to each other, as reminds

Response: Thank for your comment, we understand what comment mean, but different scales can better represent measurements and variations in each factor. We will find improved representations in future research.

figure 7: BM and AM stand for what?

Response: Thank for your comment, we added caption.

The monsoon really was the clue which have an effect on the diatom community variation, or just cases changes in environmental facts that can directly affecting the community structure?

Response: Thank for your comment, we think the large amount of rainfall on the monsoon has affected the freshwater environment facts and has caused the variation of diatom community.

Round 2

Reviewer 1 Report

The authors made corrections to the manuscript for all my comments except numbers 3 and 9.

The authors changed the name of the species Synedra acus to Tabularia arcus. But! These are different species.

Synedra acus Kützing is currently regarded as a synonym of Ulnaria acus (Kützing) Aboal. Tabularia arcus (Kützing) Williams has as a basionym Synedra arcus Kützing. That is, acus is not equal to arcus. This issue is discussed in detail in this article:

D.M. Williams & S. Blanco. Studies on type material from Kützing’s diatom collection II: Synedra acus Kützing, Synedra arcus Kützing, their morphology, types and nomenclature https://doi.org/10.1080/0269249X.2020.1711534

In order for readers to be sure that the species are identified correctly, it is desirable to bring their micrographs, you can place them in the appendix to the article.

On question number 3. For all taxa mentioned in the text, their authors should be given. Moreover, at the first mention of them. For Tabularia arcus, I have not found the author of the taxon. It is not mentioned in the text, and there is no taxon in the author's species table.

A few additional minor remarks:

1. For the authors of the article, bring ORCIDs.

2. In keywords, it is better to replace "diatom" with "diatoms".

Author Response

  1. The authors changed the name of the species Synedra acus to Tabularia arcus. But! These are different species. Synedra acus Kützing is currently regarded as a synonym of Ulnaria acus (Kützing) Aboal. Tabularia arcus (Kützing) Williams has as a basionym Synedra arcus Kützing. That is, acus is not equal to arcus. This issue is discussed in detail in this article: D.M. Williams & S. Blanco. Studies on type material from Kützing’s diatom collection II: Synedra acus Kützing, Synedra arcus Kützing, their morphology, types and nomenclature https://doi.org/10.1080/0269249X.2020.1711534. In order for readers to be sure that the species are identified correctly, it is desirable to bring their micrographs, you can place them in the appendix to the article.

Response: Thank for your comment, we carefully examined your comment. The species we identified is Synedra acus, but it is believed that there was a mistake in the process of writing the text. Synedra acus is currently accepted taxonomically as Ulnaria acus. So we revised it.

  1. On question number 3. For all taxa mentioned in the text, their authors should be given. Moreover, at the first mention of them. For Tabularia arcus, I have not found the author of the taxon. It is not mentioned in the text, and there is no taxon in the author's species table.

Response: Thank for your comment, their author of all species was mentioned in the text. Species in Table 5 but not mentioned in the text did not add their authors.

  1. For the authors of the article, bring ORCIDs.

Response: Thank for your comment

Eun-A Hwang: https://orcid.org/0000-0002-7472-8702

In-Hwan Cho: https://orcid.org/0000-0002-1945-3478

Ha-Kyung Kim: https://orcid.org/0000-0001-9279-8715

Chen Yi: https://orcid.org/0000-0002-0517-1621

Baik-Ho Kim: https://orcid.org/0000-0002-7144-0770

  1. In keywords, it is better to replace "diatom" with "diatoms".

Response: Thank for your comment, we replaced it.

Reviewer 3 Report

Some comments are not addressed properly. 

1. The authors may show some in-situ photos for Epilithic diatoms, which help reader to easily understand. It is unreasonable to consider this in the future study.

2. In addition, do you consider the distance effect, i.e., if the urban is 30 m or 50 m from the sampling place? They should have difference influences. There is no response for this issue.

Author Response

  1. The authors may show some in-situ photos for Epilithic diatoms, which help reader to easily understand. It is unreasonable to consider this in the future study.

Response: Thank for your comment, we added photo. Since this study was not a taxonomic study of diatoms, light and scanning electron microscopy images were added for species identification and cell density calculation after collection. For reference, we have been reported new diatom species recently. Please refer to it. (Diversity 2022, 14(10), 790; https://doi.org/10.3390/d14100790)

  1. In addition, do you consider the distance effect, i.e., if the urban is 30 m or 50 m from the sampling place? They should have difference influences. There is no response for this issue.

 Response: Thank for your comment, your comments are very interesting. And it is judged that it can be a very important factor in determining water quality and diatom community. On the other hand, it is not easy to select a location for sampling at a river site due to accessibility or risk. However, in future research, it is judged that the distance between the city and the gathering place must be taken into consideration. Again thanks for the nice comments

Reviewer 4 Report

I accept all of the corrections.

Author Response

Dear Reviewer,

I am sure that your kind comments has been very helpful in improving the quality of our paper. Again thanks.